# V-Set and Immunoglobulin Domain-Containing 1 (VSIG1), Predominantly Expressed in Testicular Germ Cells, Is Dispensable for Spermatogenesis and Male Fertility in Mice

**DOI:** 10.3390/ani11041037

**Published:** 2021-04-07

**Authors:** Yena Jung, Hyewon Bang, Young-Hyun Kim, Na-Eun Park, Young-Ho Park, Chaeli Park, Sang-Rae Lee, Jeong-Woong Lee, Bong-Seok Song, Ji-Su Kim, Bo-Woong Sim, Dong-Won Seol, Gabbine Wee, Sunhyung Kim, Sun-Uk Kim, Ekyune Kim

**Affiliations:** 1College of Pharmacy, Catholic University of Daegu, Gyeongsan-si 38430, Korea; icaros1019@gmail.com (Y.J.); tb642@hanmail.net (H.B.); qkrskdms3729@gmail.com (N.-E.P.); qkrcofl7863@gmail.com (C.P.); 2Futuristic Animal Resource and Research Center, Korea Research Institute of Bioscience and Biotechnology, Daejeon 28116, Korea; pyh2877@kribb.re.kr (Y.-H.P.); sbs6401@kribb.re.kr (B.-S.S.); kimjs@kribb.re.kr (J.-S.K.); embryont@kribb.re.kr (B.-W.S.); sunuk@kribb.re.kr (S.-U.K.); 3National Primate Research Center, Korea Research Institute of Bioscience and Biotechnology, Daejeon 28116, Korea; kyh@kribb.re.kr; 4Laboratory Animal Research Center, School of Medicine, Ajou University, Yeongtong-gu, Suwon 16499, Korea; lsr21@ajou.ac.kr; 5Biotherapeutics Translational Research Center, Korea Research Institute of Bioscience and Biotechnology (KRIBB), Deajeon 34141, Korea; jwlee@kribb.re.kr; 6Laboratory Animal Center, Daegu-Gyeongbuk Medical Innovation Foundation (DGMIF), Daegu 41061, Korea; seol@dgmif.re.kr (D.-W.S.); gabbine@dgmif.re.kr (G.W.); 7Department of Environmental Horticulture, University of Seoul, Seoul 02504, Korea; kimekyune@gmail.com

**Keywords:** spermatogenesis, fertility, VSIG1, fertilization, sperm

## Abstract

**Simple Summary:**

V-set and immunoglobulin domain-containing 1 (VSIG1) is a newly discovered member of the junctional adhesion molecule (JAM) family encoded by a gene located on the X chromosome in humans and mice. Expression of VSIG1 in normal mammalian tissues was originally reported to be highly tissue-specific and was detected in organs such as the testis. However, as little is known about its physiological function, we aimed to investigate the function of VSIG1 in mammalian spermatogenesis and fertilization.

**Abstract:**

To elucidate the functional role of V-set and immunoglobulin domain-containing 1 (VSIG1) in spermatogenesis and fertilization, we knocked out (KO) VSIG1 in a mouse embryo using CRISPR/Cas9 (Clustered regularly interspaced short palindromic repeat/CRISPR-associated protein 9) -mediated genome editing. Reverse transcription PCR was performed using cDNA synthesized from VSIG1 KO testis RNA. Although Western blot analysis using a specific antibody to VSIG1 confirmed VSIG1 protein defects in the KO mice, hematoxylin-eosin staining analysis was similar in the KO and wild-type mice. Additionally, computer-assisted sperm analysis and in vitro fertilization experiments were conducted to confirm the activity and fertilization ability of sperm derived from the KO mouse. Mice lacking VSIG1 were viable and had no serious developmental defects. As they got older, the KO mice showed slightly higher weight loss, male mice lacking VSIG1 had functional testes, including normal sperm number and motility, and both male and female mice lacking VSIG1 were fertile. Our results from VSIG1 KO mice suggest that VSIG1 may not play essential roles in spermatogenesis and normal testis development, function, and maintenance. VSIG1 in sperm is dispensable for spermatogenesis and male fertility in mice. As several genes are known to possess slightly different functions depending on the species, the importance and molecular mechanism of VSIG1 in tissues of other species needs further investigation.

## 1. Introduction

For fertilization to succeed in mammals, ejaculated sperm must enter the female reproductive organs where it meets an ovulated oocyte. It must then digest the cumulus cell mass around the oocyte, bind to and infiltrate the zona pellucida, and bind/fuse to the plasma membrane of the oocyte [1,2,3,4]. Despite extensive research on the molecular causes of human infertility, the underlying mechanism remains unclear. Numerous studies have shown that both male and female factors contribute to human infertility [5,6,7]. Therefore, research on the infertility mechanisms of both males and females is needed to develop the most effective treatment for infertility. Research has shown that male infertility is affected by several environmental, behavioral, and genetic factors [8]. However, research on male infertility is lacking in detail and quantity. In mammals, sperm is produced in the testes. Immature sperm enter the epididymis and gradually gain motility and fertilization ability as they pass through the male reproductive organs. Several studies have demonstrated that the interactions between the surface proteins of spermatogenic cells and Sertoli cells are important in spermatogenesis [8,9].

V-set and Ig domain-containing protein 1 (VSIG1), also known as A34, is a member of the immunoglobulin superfamily (IgSF), which is a large protein superfamily of cell surface and soluble proteins involved in cell recognition, binding, or adhesion processes [10,11,12,13]. Molecules are categorized as members of this superfamily based on shared structural features with immunoglobulins and possession of an immunoglobulin (Ig) domain or fold [14]. Members of the IgSF include cell surface antigen receptors, co-receptors, and co-stimulatory molecules of the immune system, molecules involved in antigen presentation to lymphocytes, cell adhesion molecules, certain cytokine receptors, and intracellular muscle proteins [15,16]. In 2010, a new type of IgSF, VSIG1, was identified in mouse testis. VSIG1 was first identified and characterized in humans [11]. It is predominantly expressed in the testis and is characterized by the presence of two Ig-like domains.

Despite the possible importance of VSIG1 in spermatogenesis, its function has not been well characterized. In this study, we generated VSIG1 knockout (KO) mice using the clustered regularly interspaced short palindromic repeat (CRISPR)/CRISPR-associated protein 9 (Cas9) system [17], and compared the functions of sperm from VSIG1 KO mice with those of wild-type (WT) mice. Unexpectedly, VSIG1 KO mice were fully fertile and yielded normal litter sizes. An in vitro fertilization assay demonstrated no significant difference in the functions of epididymal sperm between wild-type and VSIG1 KO mice.

## 2. Materials and Methods

### 2.1. Mouse Care

All mouse experiments were approved by the Institutional Animal Care and Use Committee of the Korea Research Institute of Bioscience and Biotechnology (KRIBB) (Approval No. (date): KRIBB-AEC-16077 (2016-04-01)) and were performed in accordance with the Guide for the Care and Use of Laboratory Animals published by the U.S. National Institutes of Health.

### 2.2. Generation of VSIG1 KO Mice

VSIG1 knock-out mouse was generated by KRIBB and mice were interbred and maintained in pathogen-free condition. Briefly, pregnant mare serum gonadotropin (PMSG) and human chorionic gonadotropin (HCG) were treated into C57BL/6N female mice. After 48 h, these female mice were mated with C57BL/6N stud male mice. Next day, virginal plug-checked female mice were sacrificed and the fertilized embryo harvested. A sgRNA and Cas9 protein mixture was microinjected into one cell embryo [18,19]. Microinjected embryos were incubated at 37 °C for 1–2 h. Fourteen to sixteen injected one-cell-staged embryos were transplanted into oviducts of pseudopregnant recipient mice (ICR). After F0 were born, genotyping using tail cut samples was performed by PCR. The primers used are listed in Table 1. PCR-positive samples were analyzed by sequencing.

### 2.3. Genomic DNA Isolation and PCR Analysis

Mice tail snips (5 mm) were digested in 1.5-mL microcentrifuge tubes (Axygen^®^) with 400-µL digestion buffer (150 mM of NaCl, 100 mM of Tris, 5 mM of EDTA, pH 8.0, 1% sodium dodecyl sulfate (SDS), and 0.2 mg of proteinase K) at 55 °C. Genomic DNA was extracted by adding 800 µL of 100% ethanol to the upper aqueous phase after adding 400 µL of phenol:chloroform (1:1) saturated with TE buffer (10 mM Tris-HCl, 0.2 mM Na_2_EDTA, pH 8.0) to the sample, and rotating thoroughly for approximately 3 h. The genomic DNA concentrations were measured using a nanodrop spectrometer (Takara, Shiga, Japan) and then adjusted to 100 ng/µL by adding nuclease-free water. To confirm the CRISPR target region in the VSIG1 gene, we designed a pair of primers (Table 1 and Appendix A) to amplify using the NCBI Primer Design Tool (ncbi.nlm.nih.gov/tools/primer-blast/, accessd on 23 February 2019) to amplify the 190-bp. The target area of *VSIG1* is shown in Figure 1 and Appendix A. For PCR, 10 µL of HiPi master-mix (Elpisbio, Daejeon, Korea) and 1 µL each of forward and reverse primers (100 µM stock; Macrogen, Seoul, Korea) were added to each PCR tube (0.2 mL; Thermo Scientific, Waltham, MA, USA). Next, 1 µL of the genomic DNA sample in 38 µL of nuclease-free water was added to each tube to a final reaction volume of 50 µL. PCR was performed in a thermocycler (Biometra T3 thermocycler, Jena, Germany) under the following conditions: Initial denaturation (94 °C for 3 min), followed by 34 cycles of denaturation (94 °C for 60 s), annealing (60 °C for 60 s), and extension (72 °C for 45 s), with a final extension at 72 °C for 4 min.

### 2.4. Preparation of Agarose Gels and Electrophoresis of PCR Products.

To confirm that CRISPR induced 2-bp deletion in VSIG1, approximately 190-bp PCR products from F1 pups were gel extracted and sub-cloned into pGEMT vectors (A3600, Promega, Madison, WI, UAS) by TA cloning and analyzed by Sanger sequencing (outsourced to Macrogen Korea). To be more specific, 4% agarose gel was prepared using low EEO agarose (A5093) of 95% purity (Sigma–Aldrich, Saint Louis, MO, USA) dissolved in 1× Tris-Acetate EDTA (TAE) buffer (40 mM Tris-acetate, 1 mM EDTA, pH 8.0) by heating the solution in a microwave oven for 3 min and it was immediately poured on a gel casting tray fitted with a 12-well comb. Then 25 µL of each PCR sample was loaded into each well and electrophoresis was performed for 2 h at 6.7 volts/cm (Optima Inc, Tokyo Japan). The TAE buffer in the electrophoresis was changed repeatedly during the operation. After electrophoresis, gel bands were observed after 10 min of reaction to 1% EtBr (ethidium bromide) solution approximately. The bands were visualized with ethidium bromide, and then excised from the gels using a razor blade. After the fragments were soaked briefly in water to remove the gel buffer and ethidium bromide, DNA purification was performed as described by Qiagen [20].

### 2.5. Total RNA Extraction and Reverse Transcription-PCR

Total RNA was obtained from VSIG1^+/+^ and VSIG1^−/−^ mouse testicular cells using Isogen, as described previously. Total RNA (5µg) was reverse-transcribed to cDNA using the SuperScript III First-Strand Synthesis System (Invitrogen, Carlsbad, CA, USA). PCR amplification was carried out using Taq DNA polymerase (Episbio, Daegeon, Korea), in accordance with the manufacturer’s instructions. One-tenth of the first strand of cDNA reaction mixture was used as the template. Specific PCR primers were designed for the amplification of the mouse genes *VSIG1* and *GAPDH*. The oligonucleotide sequences of all the primers used in this study are provided in Table 1. Their action protocol consisted of 35 cycles at 94 °C for 60 s, 60 °C for 60 s, and 72 °C for 60 s. The amplified cDNA products were analyzed using 1.5% agarose gel electrophoresis, as described previously [21].

### 2.6. Isolation of Testiscular Germ Cells (TGC) and Preparation of Protein Extracts

Testicular tissue from 4-month-old mice was minced using a razor blade in 4 mM of HEPES-NaOH, pH 7.4, containing 140 mM of NaCl, 4 mM of KCl, 10 of mM glucose, and 2 mM of MgCl_2_, filtered through a nylon mesh, and centrifuged at 1200× *g* for 10 min at 4 °C as described previously [21]. The cell pellet was suspended in the same buffer; the suspension was placed on a 52% Percoll gradient (Amersham Biosciences, Uppsala, Sweden) in the above buffer and centrifuged at 11,000× *g* for 10 min at 4 °C. TGCs were then recovered from a white band near the top of the gradient and washed three times with phosphate-buffered saline. For protein extraction, the TGCs were lysed in buffer consisting of 20 mM Tris-HCl, 1% Triton X-100, 15 mM NaCl, and 1% protease inhibitor cocktail. After centrifugation at 10,000× *g* for 10 min at 4 °C, the proteins in the supernatant solution were analyzed and the protein concentration was determined according to the Bradford method [22].

### 2.7. SDS-PAGE Electrophoresis and Western Blot Analysis

Proteins were denatured by boiling for 3 min in the presence of 1% SDS and 1% 2-mercaptoethanol, separated by SDS-polyacrylamide gel electrophoresis (PAGE), and transferred to Immobilon membranes. The membranes were blocked with Tris-buffered saline Tween-20 containing 2% skim milk, incubated with the primary anti-VSIG1 [11], anti-GAPDH (ab9485, Abcam, Cambridge, UK) at 27 °C for 2 h, and further incubated with horse radish peroxidase-conjugated goat anti-rabbit IgG for another 1.5 h. The immunoreactive proteins were detected using an enhanced chemiluminescence (ECL) Western blotting detection kit (Elpisbio, Daegeon, Korea) [23].

### 2.8. Fertility Testing

Sexually mature male WT and VSIG1^−/−^ mice were housed with female F1 progeny (>2 months old) from a C57BL/6J × DBA/2cross (also referred to as B6D2F1mice) for 4 months and the number of pups in each cage was determined within a week of their birth. Copulation was confirmed by the presence of vaginal plugs, which were checked every morning [24].

### 2.9. Epididymal Sperm Analysis with Computer-Assisted Sperm Analysis (CASA) Systems

Sperm motility and motion kinetics were measured using the CASA system (FSA 2016; Medical Supply, Seoul, Korea) [13]. Epididymal regions from either WT or VSIG1 KO mice were dissected, and the cauda part was placed in a petri dish containing 300 μL of human tubal fluid medium (HTF). Multiple incisions were made in the separated cauda to squeeze out sperm, which were then incubated at 37 °C in a 5% CO_2_ atmosphere for 1 h for capacitation. Ten microliters of sperm suspensions were loaded into a 20-μm deep Leja slide chamber (Leja, Nieuw-Vennep, the Netherlands) and sperm were analyzed for motility and concentration using CASA. In each sample, the percentage of total motile (total motility, %), progressively motile (progressive motility, %), and rapidly motile sperm (rapid motility, %), as well as the average path velocity (VAP, μm/s), curvilinear path velocity (VCL, μm/s), straight line velocity (VSL, μm/s), and amplitude of lateral head displacement (ALH, μm) of sperm, were determined. Regarding the analysis settings, sperm with straightness ≥70% and VAP ≥50 μm/s were considered progressively motile, whereas sperm with VAP ≥50 μm/s were classified as rapidly motile.

### 2.10. In Vitro Fertilization Assay

Approximately 6–7-week-old C57BL/6J female mice were super ovulated using intraperitoneal injections of 7.5 IU pregnant mare serum gonadotropin (PMSG) and human chorionic gonadotropin at 46-h intervals [12]. The cumulus-oocyte complexes (COC) were collected from the ampulla of the oviduct. Denuded oocytes were prepared by treating the COC with 0.1% hyaluronidase. The oocytes were cultured in M16 medium prior to insemination. The cauda epididymis was dissected from WT and KO mice, and gently squeezed to collect the spermatozoa in human tubal fluid (HTF) media (Origio, Måløv Denmark). The spermatozoa were capacitated at 37 °C for 1 h prior to insemination and subsequently mixed with the oocytes at a final concentration of 1.5 × 10^6^ sperm/mL. The oocytes and spermatozoa were co-incubated for 4 h at 37 °C in a 5% CO_2_ atmosphere. Cumulus-intact or cumulus-free eggs were inseminated by the capacitated sperm (1.5 × 10^5^ sperm/mL) in a 0.2-mL drop of TYH medium (i.e., a modified Krebs-Ringer bicarbonate solution supplemented with glucose, sodium pyruvate, bovine serum albumin, and antibiotics). The inseminated eggs were incubated for 6 h at 37 °C in a 5% CO_2_ environment. Following this, the cumulus cells were removed by incubating the eggs with bovine hyaluronidase (3 units/mL) for 15 min, and the eggs were subsequently washed. The female and male pronuclei of the inseminated eggs were stained with 4’-6-diamidino-2-phenylindole (10 μg/mL) for 30 min, and then viewed under a Leica fluorescence microscope.

### 2.11. Histological Analysis

Hematoxylin and eosin (HE) staining was carried out in accordance with the manufacturer’s guide (Leica Biosystems). Briefly, after fixing the testes of 5-month-old VSIG1 KO mice in Bouin’s solution (BBC Biochemical, Mount Vernon, WA, USA) for 24 h, the tissues were embedded in paraffin and cut into 4-µm-thick slices. Subsequently, morphometrical analyses were conducted using optical microscopy (DM3000, Leica, Wetzlar, Germany), following which, the slides were stained using HE staining.

### 2.12. Statistical Analysis

The data were presented as the mean ± standard error of the mean from at least three independent experiments. Statistical comparisons were made using analysis of variance, followed by a Student’s *t*-test, after the normal distribution of the data had been examined using SigmaStat Software (SPSS Inc., Chicago, IL, USA). *p*-values less than 0.05 were considered to indicate statistical significance (* *p* < 0.05).

## 3. Results

### 3.1. Establishment and Characterization of the V-Set and Immunoglobulin Domain-Containing 1 (VSIG1) KO Mouse Line

The overall strategy used to generate and analyze the VSIG1 KO mice is depicted in Figure 2. Exon 4 of VSIG1 was selected for targeting with CRISPR/Cas9 (Figure 1). As shown in Figure 1a, mouse *VSIG1* consisted of eight exons on chromosome X.

To confirm the mutation of *VSIG1*, pairs of forward and reverse primers (P1 and P2) were designed to specifically amplify the gene region targeted by CRISPR. PAGE electrophoresis (20%) indicated that the wild-type (WT) exon 4 band was negligibly high compared with the VSIG1 mutant exon 4 region (Figure 1b). As shown in Figure 1a, DNA sequencing showed that the GAC base of exon 4, which constitutes the open reading frame of *VSIG1*, was replaced with T, resulting in two base defects (Figure 1a. Therefore, a stop codon was introduced into exon 4 of reconstructed *VSIG1* (Figure 1c and Table 1). Genotyping showed that 5 of 20 founders (two males, three females) were targeted by CRISPR/Cas9 for VSIG1 KO. To confirm a line containing the deletion of the VSIG1 DNA sequence on the same homologous chromosome, the founders were backcrossed with C57BL/6J WT female mice to establish heterozygous VSIG1 KO mice. After heterozygous offspring were interbred, genomic PCR analysis followed by Sanger sequencing showed that CRISPR/Cas9 editing resulted in the elimination of four sequences in exon 4 of *VSIG1*.

To examine whether a 2-bp deletion in *VSIG1* was expressed, we produced two sense primers; one included the 2-bp deletion region (P3), whereas the other was outside of the 2-bp deletion region (P5) in the exon 4 area where DNA mutation did not occur, and an antisense primer (P4) was generated in exon 5 for reverse transcription PCR (RT-PCR) (Appendix A). Although no bands were detected in RT-PCR using the P3 and P4 primers, a band was detected using the P5 and P4 primers (Figure 3a). Finally, to determine whether the VSIG1 protein was present in the KO mice, Western blot analysis was performed on the WT and KO testicular protein extracts. VSIG1 protein was completely removed from the testes of the KO mice as shown in Figure 3b and Appendix A.

### 3.2. VSIG1 KO Male Mice Are Fertile and Have Normal Sperm Parameters

Despite the abundant expression of *VSIG1* in the mouse testes, its role in spermatogenesis remains unknown [2]. To determine whether the absence of VSIG1 affects fertility, we bred the VSIG1 KO males with fertility-proven WT females and KO females at least five times. Unexpectedly, the VSIG1 KO breeding pairs produced litter sizes comparable to the WT breeding pairs, indicating that the KO male mice were completely fertile. The normal formation of copulation plugs in mated females demonstrated that the fertilization ability of sperm from the KO mice was not impaired. Moreover, the litter sizes of the WT and KO mice were normal (average 10.7, 10.1, and 10.5 offspring, respectively) from crosses between male and female mice, respectively (Figure 4a).

Furthermore, morphological examinations of HE-stained spermatogenic cells showed that the cells in the seminiferous tubules in the testes of KO mice had a normal appearance (Figure 4b). Thus, based on these findings, we concluded that VSIG1 is not essential for mouse spermatogenesis. The progeny of the cross between VSIG1 heterozygous mice that generated the KO mice exhibited the expected Mendelian ratios (17:32:16 against wild-type; hetero-type; knock out type from six breeding pairs). Morphological analysis did not demonstrate any significant difference in the shapes of VSIG1 WT and VSIG1 KO mice. To investigate whether the absence of then VSIG1 molecule had an effect on sperm functional parameters, sperm from six-week-old KO and WT male mice were collected and analyzed by using a computer-assisted sperm analysis system (Appendix A). Similarly, cauda epididymal sperm isolated from KO mice was indistinguishable from that isolated from WT mice with respect to shape, motility, and the percentage of acrosome-reacted sperm. In addition, the average number of sperm (1.17 ± 1.15 × 10^7^ sperm/mL, 5 mice) in the cauda epididymis, and the sperm count (1.18 ± 1.75 × 10^6^ sperm/mL, 5 mice) in the uterus from the ejaculated semen of KO mice, were similar to those of WT mice (1.23 ± 2.05 × 10^7^ sperm and 1.20 ± 1.35 × 10^6^ sperm, respectively). These findings indicate that there were no apparent abnormalities in spermatogenesis, sperm maturation, or ejaculation in male KO mice.

### 3.3. VSIG1 KO Male Mice Exhibit Normal Fertility in IVF

To assess the VSIG1 KO mouse sperm and egg interaction, an in vitro fertilization assay was carried out using capacitated cauda epididymal sperm. When cumulus-intact eggs were used, the fertilization rate after insemination with VSIG1 KO mouse sperm was normal (Figure 5a). No significant difference was found either in the sperm binding to zona-pellucida (ZP) or in the fusion of sperm with ZP-free eggs between WT and KO mice (Figure 5b).

## 4. Discussion

Many testis-specific proteins required for spermatogenesis are suspected to play a role in male fertility. Elucidating the functions of these proteins may improve genetic diagnosis of infertility, as well as the ability to select optimal therapeutic strategies. Spermatogenesis is a complex process controlled by pituitary adenylate cyclase-activating peptide in conjunction with local testicular factors, including testicular steroidogenic enzymes [25,26,27]. Gene knockout (KO) is used to determine impact of the absence of one or more genes on in vivo functioning. Of additional utility, KO of genes exhibiting organ-restricted expression patterns (e.g., testis-specific genes) often does not affect organism viability. The present study determined the function of VSIG1, a testicular germ cell surface transmembrane protein, using KO mice generated by means of CRISPR/Cas9 technology. Specifically, replacement of three bases (GAC) with T resulted in formation of a stop-codon, leading to the expression of a truncated version of VSIG1. Unexpectedly, findings demonstrated that VSIG1 is not essential for spermatogenesis or fertilization. Additionally, RT-PCR demonstrated no significant impact of mutation on mRNA transcription, even when a 2-bp deletion within exon 4 resulted in formation of a stop codon, also leading to VSIG1 truncation. However, Western blotting confirmed the absence of both full-length and truncated VSIG1 in the VSIG1 KO testis. Counter to expectations, spermatozoa of VSIG1 KO mice did not differ significantly from those of wild-type mice in terms of morphology, counts, and function.

Spermatogenesis occurs within the seminiferous tubules of the testis, and interaction of spermatogenic cells with Sertoli cells is important for this process. Proteins (e.g., Ctx, Car, Bt-Igsf, and Jam-c) belonging to the same family as VSIG1 are essential for spermatogenesis, making the finding that VSIG1 is dispensable for spermatogenesis all the more unexpected. One cause of male infertility is a lack of certain proteins required for spermatogenesis. Testis-specific proteins ADAM1a (A Disintegrin And Metalloprotease) and CLGN (Calmegin) are responsible for transporting newly-synthesized cell surface proteins, such ADAM3, to the cell membrane during spermatogenesis [28,29,30]. Although these proteins themselves are not present in wild-type spermatozoa, ADAM1a or CLGN KO male mice have been demonstrated to be infertile, and during in vitro fertilization (IVF) such spermatozoa exhibited poor zona pellucida (ZP) binding ability. The present study therefore also sought differences during IVF between VSIG1 KO and wild-type spermatozoa. However, no significant differences were present. Taken together, these findings demonstrate that VSIG1 is dispensable for murine male fertility and does not play critical roles during in vitro or in vivo murine fertilization. Indeed, recent gene KO studies have demonstrated that many fertilization factors are dispensable for fertility, likely due to the presence of multiple redundant mechanisms. This may be why no obvious phenotypic difference has been observed in the majority of other testis-specific gene (e.g., claudin, Cxadr (coxsackievirus and adenovirus receptor)) KO mice [31,32,33]. An additional possible explanation is genetic redundancy: Genes from similar types of gene families can play common roles, facilitating continued functioning even if one gene is deleted. One example involves the ability of *HYAL5* to compensate for deletion of *SPAM1*, a spermatozoan surface hyaluronidase, and vice versa. However, when these two genes are simultaneously deleted, fertilization is critically impaired [12,34,35,36].

## 5. Conclusions

Our findings demonstrate that VSIG1 protein is non-essential for spermatogenesis and does not fulfill a critical role in in vitro fertilization in mice. As revealed by various studies, many genes may have slightly different functions depending on the species; thus, the importance of VSIG1 in other species needs to be further studied. Regardless, the molecular mechanism of VSIG1 in other tissues remains to be further elucidated.

## Figures and Tables

**Figure 1 animals-11-01037-f001:**
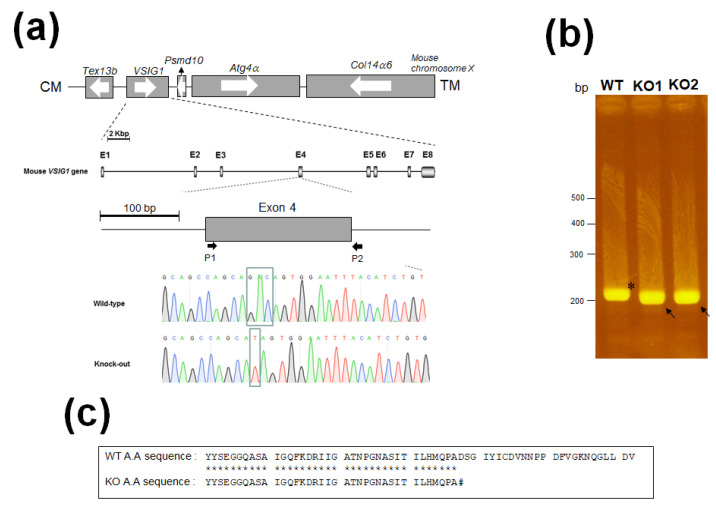
Generation of V-set and immunoglobulin domain-containing 1 (VSIG1)^−/−^ mice using clustered regularly interspaced short palindromic repeat/CRISPR-associated protein 9 (CRISPR/Cas9) system. (**a**) Mouse VSIG1 contains eight exons (shaded boxes) interrupted by three introns. The ATG translation initiation codon is located in the second exon. Genomic sequence of the CRISPR target site in VSIG1^+/+^ and VSIG1^−/−^. Letters in the box indicate the target mutation. (**b**) Genomic DNA PCR from KO mice. Arrowhead means wild-type (WT) band, while arrow means VSIG1 knock out (KO) of male and female. (**c**) Comparison of peptide sequences between wild-type (WT) and knock out (KO) VSIG1 mice. Amino acids identical between two sequences are shown asterisks. # means stop cordon.

**Figure 2 animals-11-01037-f002:**
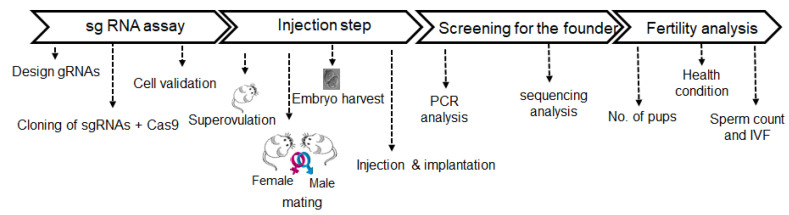
Overall strategy for the design, generation, and functional analysis of VSIG1.

**Figure 3 animals-11-01037-f003:**
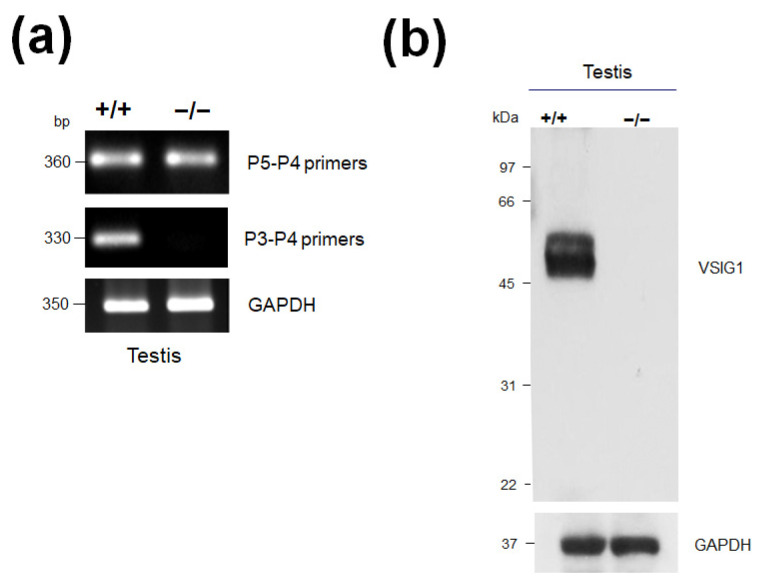
(**a**) RT-PCR analyses of total RNA from the testes of *VSIG1^+/+^* and *VSIG1^−/−^* mice. The fragments were separated by 1.5% agarose gel electrophoresis. (**b**) Western blot analysis of protein extracts from the testes of the *VSIG1^+/+^* and *VSIG1^−/−^* mice, using an affinity-purified anti-VSIG1 antibody. No immunoreactive protein corresponding to the 55 kDa VSIG1 was found in the testes of the *VSIG1^−/−^* mice.

**Figure 4 animals-11-01037-f004:**
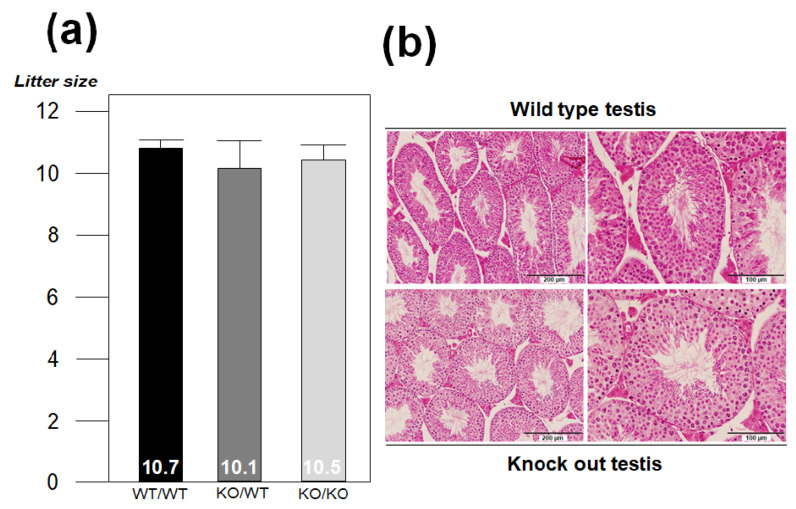
The loss of VSIG1 does not affect spermatogenesis or the rate of fertilization. (**a**) Mean litter size from VSIG1 KO and WT male mice. (**b**) Hematoxylin and eosin staining of the mouse testis in wild-type (WT) and VSIG1 knockout (KO) mice.

**Figure 5 animals-11-01037-f005:**
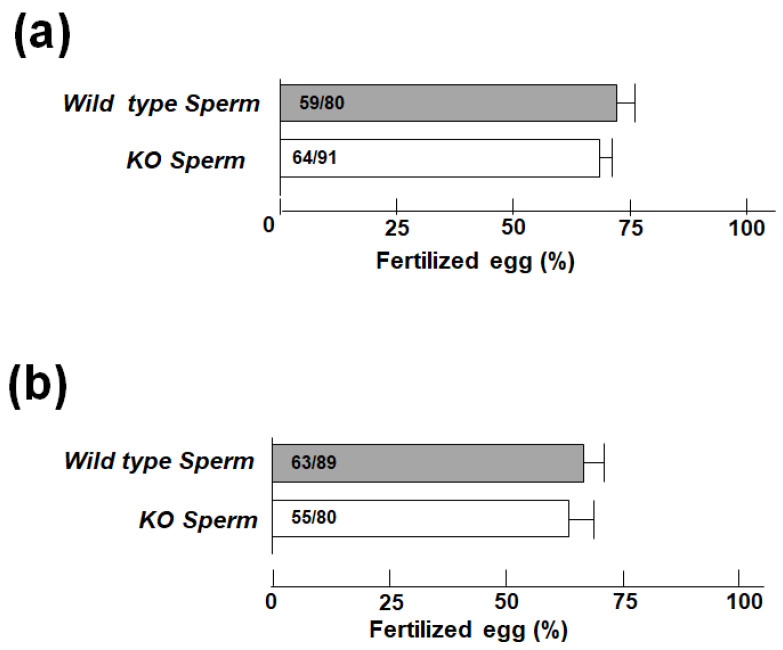
In vitro fertilization (IVF) assay using VSIG1 knockout (KO) mouse sperm. (**a**) The fertilization ratio of intact eggs inseminated with capacitated cauda epididymal sperm from VSIG1 KO and wild-type (WT) male mice. (**b**) Cumulus-free eggs were inseminated with capacitated cauda epididymal sperm derived from WT and VSIG1 KO mice, and the fertilization ratio was determined.

**Table 1 animals-11-01037-t001:** Number of primer sequences for genomic PCR (polymerase chain reaction) and RT-PCR (reverse transcription PCR).

	Sense Primer (5′to 3′)	Anti Sense Primer (5′to 3′)
VSIG1 gPCR	P1: TACTACTCTGAAGGTGGACAG	P2: GTTTGACTAAGACAGTGACGAC
VSIG1 rtPCR	P3: TATTGCATATGCAGCCAGCAGAC P5: GGCTACTAATCCCGGTAATGCAT	P4: GTACACTGGTAACCTTGTTC
GAPDH rtPCR	AGATTGTCAGCAATGCATCCTG	TGCTTCACCACCTTCTTGATGT

gPCR; genomic polymerase chain reaction, rtPCR; reverse transcription polymerase chain reaction.

## Data Availability

Not applicable.

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
