# Peer review of "V-Set and Immunoglobulin Domain-Containing 1 (VSIG1), Predominantly Expressed in Testicular Germ Cells, Is Dispensable for Spermatogenesis and Male Fertility in Mice"

_animals, 2021, doi:10.3390/ani11041037_

Round 1

Reviewer 1 Report

The authors made all the proposed changes. I think the quality of the manuscript has improved whereby it can be accepted for publication in Animals.

Reviewer 2 Report

The revised version looks better now. I think it is considered to be acceptable. Thank you!  

This manuscript is a resubmission of an earlier submission. The following is a list of the peer review reports and author responses from that submission.

Round 1

Reviewer 1 Report

Jung and colleagues aimed to settle if V-set and immunoglobulin domain containing (VSIG1; a protein highly expressed in stomach and testis) has roles in spermatogenesis and sperm function. In order to do that, they claim they have created VSIG1-KO mice using CRISPR/Cas. Since no differences on sperm fertilization ability were observed between KO and wild type mice, the authors concluded VSIG1 is not important for male fertility in mice.

Overall, the work has several flaws, the manuscript is not well organized and technical details are missing. I have nothing against negative data, but I believe the data shown is not enough to justify publication. 

Some specific points:

- The first part of the Results section (VSIG1-KO strategy) should be in Material and Methods.

- Technical details missing include (among others): guideRNAs used, the age of the mice used; how were mouse testes isolated/prepared; culture media constitution; CASA setup; how were sperm capacitated; etc

- Line 195: how come the fact that there is no difference in band size indicates a successful KO?

- Which regions primers P3/P5 and P4 flank? The outcomes in Fig.3A are not clear. Fig.3B should show the entire blot and more samples.

- Was SigmaStat software used for statistics (as indicated in the legend of Fig.5) or Excel (as indicated in Material and Methods)? And how was the normality of data tested?

- Fig.5: graphs have no units.

- Line 254-255: how was the percentage of acrosome-reacted sperm assessed? How was the sperm count in the uterus determined?

- Fig 6: what was the reasoning for using cumulus-free eggs on the second part of the experiment?

Author Response

Dear reviewer

Thank you for thinking positively about our manuscript. As Reviewer pointed out, we could find our data had mistakes. The reviewer's suggestion helped to improve the quality of the paper. In fact, it took a lot of time to make VSIG1 KO mice. Contrary to expectations, however, the VSGI 1 male KO mouse had normal ability to modify. Nevertheless, we are proud that VSIG1 is being announced by our results, and we hope you will think positively. Below are the amendments pointed out by the reviewer.

Some specific points:

- The first part of the Results section (VSIG1-KO strategy) should be in Material and Methods.

Answer: As suggested by the reviewer, the first part of the result was moved to Material and Methods part. And I wrote it in more detail.

- Technical details missing include (among others): guideRNAs used, the age of the mice used; how were mouse testes isolated/prepared; culture media constitution; CASA setup; how were sperm capacitated; etc

Answer: Eight-week-old male KO mouse was used in the CASA experiment. The spermatozoa were capacitated at 37°C for 1 h prior to CASA and IVF. Then, the experiment method was described in detail. Please refer to the materials and methods.

- Line 195: how come the fact that there is no difference in band size indicates a successful KO?

Answer: The reviewer made absolutely the right point. We carried out 1.5% agarose gel as products of Genomic PCR but could not confirm the difference in band size. So, we were able to check the difference in band by using 20% PAGE gel for 5 hours’ electrophoresis, and after extracting DNA from two bands and carrying out TA cloning, we proceeded sequencing to confirm that two bases were deleted. So, I removed "indicates a successful KO" from the sentence.

- Which regions primers P3/P5 and P4 flank? The outcomes in Fig.3a are not clear. Fig.3b should show the entire blot and more samples.

Answer: The sentences we write can be hard to understand. The picture description of Fig3 was detailed, and the figure for the Western result of Fig3b was attached again

- Was SigmaStat software used for statistics (as indicated in the legend of Fig.5) or Excel (as indicated in Material and Methods)? And how was the normality of data tested?

Answer: SigmaStat software used for statistics in present research. So, we modified the legend of Fig. 5.

- Fig.5: graphs have no units.

Answer: We made a mistake writing unit on figure5 by mistake. So, I put a "gram" on the y-axis

- Line 254-255: how was the percentage of acrosome-reacted sperm assessed? How was the sperm count in the uterus determined?

Answer: Acrosome reaction evaluation was not conducted separately.

Mating experiment was conducted at 8 p.m. and it was confirmed we mating was done through copulatory plug checking in the female’s vagina the next morning. Of course, WT female was used in the experiment after injection of gonadotropin and human chorionic gonadotropin.

- Fig 6: what was the reasoning for using cumulus-free eggs on the second part of the experiment?

Answer: When certain proteins are not produced in the sperm formation process, infertility may be caused by the absence of certain membrane proteins on the mature sperm surface. Therefore, IVF was carried out with and without COC.

Reviewer 2 Report

The authors demonstrated that the lack of VSIG1 expression, a protein involved in the interaction between germ cells in the testis, does not lead to a reduction of male fertility. These data suggested to the authors the presence of redundant molecules that compensate for the lack of VSIG1

Although the data are preliminary, this is an interesting study and is a potentially valuable addition to the literature related to the control of male reproduction in vertebrates

The claims represent a novelty in the literature of vertebrate reproduction. The claims are convincing. The paper is complete. The hypotheses provided by the authors are correctly discussed in the context of the literature on the fertility of vertebrates.

This manuscript describes the relationship between VSIG1, a member of the 63 immunoglobulin (IgSF) superfamily, expressed in testicular cells, and male infertility in mice. In particular, the authors demonstrated that in VSIG1 knockout (KO) mice, there are no alterations in both spermatogenesis and in the ability sperm fertility.  Although the data are preliminary, this is an interesting study and is a potentially valuable addition to the literature related to the control of male reproduction in vertebrates. Further revisions should be done to improve the quality of the manuscript before it can be accepted for publication.

-Line 278: Authors should add that different molecules (neuropeptides and sex hormones) are involved in male fertility. In support of these data, the authors should cite some manuscripts:

Prisco, M., Rosati, L., Agnese, M., Aceto, S., Andreuccetti, P., Valiante, S., 2019. Pituitary adenylate cyclase-activating polypeptide in the testis of the quail Coturnix coturnix: Expression, localization, and phylogenetic analysis. Evolution and Development 21, 145-156.

Prisco, M., Rosati, L., Morgillo, E., Mollica, M.P., Agnese, M., Andreuccetti, P., Valiante, S., 2020. Pituitary adenylate cyclase-activating peptide (PACAP) and its receptors in Mus musculus testis. General and Comparative Endocrinology 15, 286:113297

Rosati, L., Di Fiore, M.M., Prisco, M., Di Giacomo Russo, F., Venditti, M., Andreuccetti, P., Chieffi Baccari, G., Santillo, A., 2019. Seasonal expression and cellular distribution of star and steroidogenic enzymes in quail testis. Journal of Experimental Zoology part B Molecular and Developmental Evolution 332, 198-209.

-Figure 4b: the images must be changed with better resolutions

Author Response

Dear reviewer

Thank you for thinking positively about our manuscript. As Reviewer pointed out, we could find our data had mistakes. The reviewer's suggestion helped to improve the quality of the paper. In fact, it took a lot of time to make VSIG1 KO mice. Contrary to expectations, however, the VSGI 1 male KO mouse had normal ability to modify. Nevertheless, we are proud to confirm VSIG1 function, and we hope you will think positively. Below are the amendments pointed out by the reviewer.

-Line 278: Authors should add that different molecules (neuropeptides and sex hormones) are involved in male fertility. In support of these data, the authors should cite some manuscripts:

Answer: We inserted a sentence in the introduction part of the discussion that neuropeptides and sex hormones are involved in sperm formation and attached a reference.

Prisco, M., Rosati, L., Agnese, M., Aceto, S., Andreuccetti, P., Valiante, S., 2019. Pituitary adenylate cyclase-activating polypeptide in the testis of the quail Coturnix coturnix: Expression, localization, and phylogenetic analysis. Evolution and Development 21, 145-156.

Prisco, M., Rosati, L., Morgillo, E., Mollica, M.P., Agnese, M., Andreuccetti, P., Valiante, S., 2020. Pituitary adenylate cyclase-activating peptide (PACAP) and its receptors in Mus musculus testis. General and Comparative Endocrinology 15, 286:113297

Rosati, L., Di Fiore, M.M., Prisco, M., Di Giacomo Russo, F., Venditti, M., Andreuccetti, P., Chieffi Baccari, G., Santillo, A., 2019. Seasonal expression and cellular distribution of star and steroidogenic enzymes in quail testis. Journal of Experimental Zoology part B Molecular and Developmental Evolution 332, 198-209.

-Figure 4b: the images must be changed with better resolutions

Answer: We tried to detect it under a new microscope. So, I changed the images.

Reviewer 3 Report

Manuscript: V-set and immunoglobulin domain containing (VSIG1), predominantly expressed in testicular germ cells, is dispensable for spermatogenesis and male fertility in mice.

In the current manuscript, authors investigated whether VSIG1 affected mammalian spermatogenesis and fertilization. They knocked out (KO) VSIG1 in a mouse embryo using CRISPR/Cas9-mediated genome editing. However, the KO mice failed to present differences in spermatogenesis and male fertility compared with WT mice. Overall, this work explored the function of VSIG1 in spermatogenesis and male fertility. Some concerns should be addressed before publication.

Major concerns:

  1. Are the truncated VSIG1 (from exon2 to exon4) expressed in KO mice are still functional?
  2. Could you check VSIG1 expression in sperms?
  3. You may put commercial VSIG1 KO mouse (https://www.modelorg.com/en/mice/6462/ post_ type/3.html) as a positive control in your experiment. Or You may directly use that mouse for functional analysis.
  4. In Figure 2A, WT and KO bands failed to show differences. Could you check out this paper (https://www.nature.com/articles/s41598-019-39950-4) to make WT and KO bands different?

Minor concerns:

1 Please label clearly in Figure 2B with P3, P4, P5 amplified direction. And in Figure 3A, and brand clearly with P3/P4 and P5/P4 products.

2 To make the same abbreviated words for KO. For example, K/O in Figure5.

Author Response

Dear reviewer

Thank you for thinking positively about our manuscript. As Reviewer pointed out, we could find our data had mistakes. The reviewer's suggestion helped to improve the quality of the paper. In fact, it took a lot of time to make VSIG1 KO mice. Contrary to expectations, however, the VSGI 1 male KO mouse had normal ability to modify. Nevertheless, we are proud to confirm VSIG1 function, and we hope you will think positively. Below are the amendments pointed out by the reviewer.

Major concerns:

  1. Are the truncated VSIG1 (from exon2 to exon4) expressed in KO mice are still functional?

Answer: When we carried out Western using an antibody recognizing the extracellular domain of VSIG1 protein, the absence of a band in VSIG1 knock out testis confirmed that truncated VSIG1 also did not exist.

  1. Could you check VSIG1 expression in sperms?

Answer: Yes, we first produced antibodies against VSIG1 and then proceeded with Western Blots using mouse tissues including testis and sperm. The Result showed that VSIG1 protein did not exist in Sperm. We attach a result of the Western blot experimented with the mouse tissues and sperm in wild type (previously data).

  1. You may put commercial VSIG1 KO mouse (https://www.modelorg.com/en/mice/6462/ post_ type/3.html) as a positive control in your experiment. Or you may directly use that mouse for functional analysis.

Answer: The VSIG1 knock out mouse used in the present experiment was produced by Dr. Jung Woong Lee, a co-author. The method of manufacture is described in detail in Materials and Methods.

  1. In Figure 2A, WT and KO bands failed to show differences. Could you check out this paper (https://www.nature.com/articles/s41598-019-39950-4) to make WT and KO bands different?

Answer: The PCR was carried out by applying the method proposed by the reviewer. In other words, we conducted electroporation with 20% PAGE gel to identify the difference in the band, and after TA-cloning for each band to conduct sequencing. We attach the result.

Minor concerns:

1 Please label clearly in Figure 2B with P3, P4, P5 amplified direction. And in Figure 3A, and brand clearly with P3/P4 and P5/P4 products.

Answer: As the reviewer pointed out, primer explanation was insufficient. We described it in more detail. Please confirm supplemental figure 2.

2 To make the same abbreviated words for KO. For example, K/O in Figure5.

Answer: As suggested by the reviewer, K/O of Figure 5 was replaced with KO.